# The State of the Art of Data Mining Algorithms for Predicting the COVID-19 Pandemic

Keila Vasthi Cortés-Martínez, Hugo Estrada-Esquivel *, Alicia Martínez-Rebollar *, Yasmín Hernández-Pérez * and Javier Ortiz-Hernández

TECNM/Centro Nacional de Investigación y Desarrollo Tecnológico, Cuernavaca 62490, Mexico;
d16ce003@cenidet.tecnm.mx (K.V.C.-M.); javier.oh@cenidet.tecnm.mx (J.O.-H.)
* Correspondence: hugo.ee@cenidet.tecnm.mx (H.E.-E.); alicia.mr@cenidet.tecnm.mx (A.M.-R.);
yasmin.hp@cenidet.tecnm.mx (Y.H.-P.)

**Abstract:** Current computer systems are accumulating huge amounts of information in several application domains. The outbreak of COVID-19 has increased rekindled interest in the use of data mining techniques for the analysis of factors that are related to the emergence of an epidemic. Data mining techniques are being used in the analysis and interpretation of information, which helps in the discovery of patterns, planning of isolation policies, and even predicting the speed of proliferation of contagion in a viral disease such as COVID-19. This research provides a comprehensive study of various data mining algorithms that are used in conjunction with epidemiological prediction models. The document considers that there is an opportunity to improve or develop tools that offer an accurate prognosis in the management of viral diseases through the use of data mining tools, based on a comparative study of 35 research papers.

**Keywords:** data mining; pandemic; machine learning; predictive epidemiological models

**MSC:** vector calculus; statistics; logarithmic functions

## 1. Introduction

Currently, several research groups, academic institutions, hospitals, and consulting companies have developed predictive and prognostic models for the enormous health problem represented by the COVID-19 pandemic. The purpose of those research efforts is to give support to health systems in making strategic decisions, in planning measures to control the spread of the virus, in defining social isolation plans that limit the possibilities of contagions, as well as in the formulation of pandemic mitigation measures. Every day, medical information systems are producing enormous amounts of data which are impossible for researchers to manually analyzed. Given the current needs created by a health emergency, data mining in conjunction with machine learning is a key aspect in promoting the development of epidemiological prediction models that emulate the evolution and behavior of COVID-19 [1].

One of the objectives of epidemiological models is to measure the spread of infectious disease, defining factors that make it transit from endemic to the epidemic and finally, to pandemic [2,3], remembering the task of characterizing the dynamics of a virus usually becomes complex when deciphering knowledge with predictive purposes. In relation to this, data mining plays an important role in the discovery and extraction of unknown patterns [4], this being a key aspect in the creation of models that define the mechanics behind the spread of a disease, which implies making use of algorithms that have roots in machine learning [5].

Current scientific literature indicates that since COVID-19 is a recent pandemic, the large volume of raw data from health or medical care is usually voluminous and varied and lacking in organization and integration in real-time. This, could lead to uncertainty

in the analysis of the pandemic. Therefore, the need arises to understand which tools can identify behavior patterns, help decision-making, and obtain trends with reliable results to predict future scenarios in the evolution of the disease.

The relevance of data mining techniques in the context of COVID could be explained with their ability to find hidden patterns using large databases for a very new disease without previous knowledge about its evolution in time. In the case of COVID data mining data mining techniques are useful to extract meaningful information from complex raw data providing multiple benefits in the healthcare sector [6]. The work of Safdari [7] also mentioned the significant impact of data mining techniques on selecting the most effective techniques in pandemic studies by helping the researchers to reveal the unknown character of a new pandemic and the next possible pandemics.

This paper aims to highlight the relevance of the integration of mathematical epidemiological models with data mining techniques and/or machine learning algorithms in predicting the evolution of the COVID-19 pandemic. The state-of-the-art review covers 32 research articles, seven books, and three surveys that one related to this topic.

The rest of this work is organized as follows: Section 2 present a related work section. Section 3 shows the main mathematical epidemiological models used in the analysis of COVID-19. Section 4 details the state of the art in predicting pandemics using data mining algorithms. Section 5 shows a discussion of the work presented. Finally, Section 6 presents conclusions and future work.

## 2. Related Works

The pandemic of COVID-19 has pointed out the relevance of mathematical models and data mining techniques to predict the evolution of diseases. In this context some research groups have performed reviews to analyze papers that use data mining techniques for prediction activities in pandemics such as the case of COVID-19.

The work of Clement [8] reviews the state-of-the art mathematical models for COVID-19, including compartment models, statistical models and machine learning models to provide more insight, so that an appropriate model can be well adopted for the disease spread analysis. Authors established that accurate diagnose of COVID-19 is an essential process to identify the infected person and control further spreading. A comprehensive review on the deep learning models for the diagnosis of the disease is also provided in this paper. The main difference between the Clement paper and the research work presented in this paper is that in our case, the literature review is focus on finding mathematical models and data mining techniques that extend or to improve the current epidemiological model proposed by health organizations. In case of Clement, they analyzed alternative techniques such as the use of natural language processing for the analysis of mental illness and feeling of the public based on the comments posted in the social media

The works of Mehrotra [6] reviews the various papers published on COVID-19 using data mining techniques to address the pandemic in terms of its explanation, assessment and solution. The paper analyzes the role of data mining techniques in classification and regression, text analysis and NLP, clustering, and association. The current paper studied a few most frequently cited papers and pointed out the fact that various data mining techniques were used for studying different types of issues associated with COVID-19 since prevention mechanisms to solution finding. The main difference between Mehrotra and the research work presented in this paper is that Menrotra consider a broad scope of papers, for example for sentiment analysis, topic modeling, pandemic solutions, virus spread, forecasting, etc. Our paper is focus on the specific task of prediction of the pandemic evolution in which the basis are the epidemiological models provided by Health organizations.

The work of Safdari [7] studies the published articles to determine the most favorite data mining methods used in research centers to fight with current pandemics. 50 articles were determined as eligible articles through a scoping review. The research work has classified the method into 14 categories. The review results showed that the most favorite data mining technique is Natural Language Processing (22% of reviewed papers) and the

most commonly proposed approach was revealing disease characteristics (22% of reviewed papers. The study reveals that most addressed disease was COVID-19. Safdari paper has found that most common software used in the studies was SPSS (22%) and R (20%). The main difference between Safdari and the research work presented in this paper is that the former is that consider papers in a widely scope considering by example, treatments, tracing transmission, prevention, patient monitoring, prediction, early diagnosis or active case prediction. Our paper is focus on the specific task of prediction of the pandemic evolution in which the basis are the epidemiological models provided by Health organizations.

## 3. Mathematical Epidemiological Models Used in the Prediction of COVID-19

Epidemiological models are the result of joining mathematical modeling and epidemiology, and their main objective is the study of the spread of diseases [9]. These models provide useful information to determine the prevalence and incidence of infectious diseases, serve as a tool to help in making objective decisions for the control or suppression of these diseases, and even help to prevent future contagion.

In 1927, W. O. Kermack and A. G. McKendrick [10] developed the SIR model (susceptible, infected and recovered) as the simplest model that is capable of capturing the main characteristics of epidemic outbreaks.

It is important to point out that not all diseases spread in the same way. Therefore, the SIR model is the basis for the development of several types of mathematical epidemiological models (Table 1), such as the SIS model (Susceptible, Infected and Susceptible), and the SEIR model (Susceptible, Exposed, Infected and Recovered). These models attempt to identify the dynamics in which a disease evolves by using historical data (starting from the location in which a person is susceptible to being infected) and also to model the process in the evolution in the contagion of an epidemiological disease [11], the equilibrium points of the disease [12,13] and the corresponding equilibrium in the epidemiological systems [12,14,15].

**Table 1.** Definition of Josu [13] and Rovira [16] for the main epidemiological models.

| Model | Estates | Definition | Mathematical Equations | Scheme |
|-------|---------|------------|------------------------|--------|
| SIS | S (Susceptible) I (Infected) | A susceptible $S(t)$, when passing to the infected $I(t)$ group, can die as a result of the disease or naturally, but births are considered over time, thus there is a renewal of individuals susceptible to the disease, that is, the number of infected people never reaches the total population N, a balance is maintained between infected individuals with those who are susceptible. | $\frac{dS}{dt} = -\beta SI\, \mu(N-S) + \gamma I$ $\frac{dI}{dt} = \beta SI - \gamma I - \mu I$ |  |
| SIR | S (Susceptible) I (Infected) R (Recovered) | Individuals start out as susceptible $S(t)$ to a given pathogen and, if they become infected $I(t)\ R(t)$, they progress to the other two states. (1) | $\frac{dS}{dt} = -\beta SI\, \mu(N-S)$ $\frac{dI}{dt} = \beta SI - \gamma I - \mu I$ $\frac{dR}{dt} = \gamma I - \mu R$ |  |

**Table 1.** *Cont.*

| Model | Estates | Definition | Mathematical Equations | Scheme |
|---|---|---|---|---|
| SEIR | S (Susceptible) E (Exposed) I (Infected) R(Recovered) | The exposed *E (t)* state is added, in which the disease has a long latency or incubation period, during which it may or may not infect *I (t)* other, die or recover *R (t)*. | $\frac{dS}{dt} = B - \beta SI\ \mu S$ $\frac{dE}{dt} = \beta SI - (\varepsilon + \mu) + E$ $\frac{dI}{dt} = \varepsilon E - (\gamma + \mu)I$ $\frac{dR}{dt} = \gamma I - \mu R$ |  |

*S*: Susceptible individuals.
*E*: Exposed and in latency individuals.
*I*: Infected individuals.
*R*: Recovered individuals with immunity.
*N*: Total population.

$t$: Time
$\frac{1}{\gamma}$: Average time of infections (for a single individual).
$\frac{1}{\varepsilon}$: Average incubation time.
$\beta$: Infection rate (probability that a person becomes ill when in contact with an infected person).
$\mu$: Average death rate (probability that an infected individual will die from disease).

The advantage of using mathematical models is that it is possible to modify the parameters of a system to represent or discover situations that are difficult to reproduce in a real environment. There are two types of mathematical models that are relevant for discovering knowledge in health environments: (a) deterministic, where the factors involved in the process can be controlled and the results can be accurately predicted, and (b) stochastic, where the factors that are involved in the study of the phenomenon cannot be controlled and there are several possible results. In both types of models, it is assumed that the interaction between the individuals involved in a pandemic is random [2].

The stability of an epidemiological model can be defined by its equilibrium points, which are usually defined in terms of the Basic Reproduction Number ($R_0$). Following, we define the concepts of equilibrium in epidemiological systems and the basic reproduction number:

- Equilibrium epidemiological systems. One of the important objectives in the study of a disease is to know if it will persist in a specific population, or if this disease gradually disappears until its eradication [9]. There are two types of equilibrium: Disease-free and Endemic. *Disease-free Equilibrium*: In this type, the proportion of exposed and infectious individuals is zero so that the disease is eradicated [9,12], that is, this situation occurs when the infectious subpopulations are canceled [13]. *Endemic Equilibrium*: In this type, the proportion of exposed and infectious individuals is not zero [14]; however, the capacity of the system is stable enough to remain in equilibrium even with small disturbances. These small disturbances cause the epidemiological models to begin to evolve away from a state of infection [9]. In this case, the disease cannot be totally eradicated but remains in the population. The equilibrium in epidemiological systems has been thoroughly analyzed in the research works [12,14,15] where they simulate preventive measures, containment, and the possible impact of a pandemic in the population.
- Basic Reproduction Number. In an epidemic, a very important parameter is the Basic Reproduction Number ($R_0$), it allows us to distinguish between an epidemic state (when the pathological process exceeds the expected contagions and the geographical limits) and an endemic state (when the pathological process remains stationary for an extended period of time). The Basic Reproduction Number is essential to be able

to understand the nature of the different diseases and their temporal evolutions [13]. The ($R_0$) is defined as the mean number of secondary infections that occur when an infectious individual is introduced into a susceptible population. It can be represented as $R_0 = \frac{\beta}{\gamma}$ [9], that is, the number of individuals to be infected from patient zero. In order to eradicate an infection, it is necessary to reduce the $R_0$ below unity [13]. This is sometimes accomplished through immunization programs, which have the effect of transferring members of the susceptible class (S) to the recovered class (R). The smaller the $R_0$, the slower the epidemic will evolve. In practice, for a specific real epidemic, observing the epidemic allows us to measure $R_0$ and, them estimate $\beta$ [11].

## 4. The State of the Art for Predicting the COVID-19 Pandemic Using Data Mining Algorithms

In this section, 32 articles, seven books, and three surveys, that are related to the COVID-19 pandemic are analyzed. The research works are categorized as follows:

1.　Prediction with epidemiological models using mathematical algorithms
2.　Prediction with machine learning and/or data mining algorithms
3.　COVID-19 Datasets.

The details of each of these categories are presented below.

### 4.1. Approach Used for the Papers Selection

It is important to point out that current version of paper of the paper doesn't pretend to be a systematic review of the literature of data mining techniques applied in the prediction of COVID-19 evolution. This paper has been defined as a preliminary scoping search of relevant works. However some of the steps of a systematic review has been followed in out paper:

The questions to answer in this search were: What are the different data mining techniques that have been used to predict the behavior of the COVID-19? What specific epidemiological model is used in that papers? What are the metrics used to measure the precision of the proposed data mining techniques? What are the data sources of the analyzed research works?

The inclusion criteria was to consider all works using mathematical algorithms and data mining techniques combined with a specific epidemiological model (SEIR, SIR, eSIR, etc) to intend to predict the pandemic behavior of COVID-19.

The exclusion criteria was to eliminate all works using data mining techniques for other not pandemic diseases and also eliminate of the review all papers that do not have the basis of an epidemiological model (SEIR, SIR, eSIR, etc.) to be combined with methodological models or data mining techniques.

This inclusion and exclusion criteria are the reason why this paper does not consider the use of alternative techniques to detect/predict the COVID-19, such as sentiments analysis based on natural language processing techniques, images recognition for COVID-19 detection, use of sensor for preliminary diagnosis, etc. This paper is focus on mathematical models and data mining techniques that extend/improve the current epidemiological model proposed by health organizations.

Following, we present the research works for predicting COVID-19 with epidemiological models using mathematical algorithms and the research work for predicting COVID-19 with machine learning and/or data mining algorithms.

### 4.2. Prediction with Epidemiological Models Using Mathematical Algorithms

Currently, epidemiological projection through mathematical models is of vital importance for the current COVID-19 pandemic. For example, some research works focus on locating the outbreak where a pandemic originates studying the transmission of the virus in the short and medium-term. The SIR model is used in this situation, because during the initial period of disease outbreaks, data is scarce and full of uncertainty [17,18]. Table 2 presents research works that combine mathematical approaches with the epidemiological SIR model.

**Table 2.** Mathematical algorithms combined with SIR epidemiological model.

| Reference | Country | Task Performed | Epidemiological Mathematical Model | Mathematical Prediction Algorithms |
|---|---|---|---|---|
| [3] | India | Evaluate the effectiveness of protocols | eSIR | - Bayesian and Poisson regression model<br>- Markov-Chain Monte Carlo (MCMC) |
| [14] | Pakistan | Predict Growth and Transmission | SIR | - Lipschitz condition<br>- Disease Free Equilibrium (DFE)<br>- Basic Reproduction Number ($R_0$)<br>- Jacobian<br>- Lyapunov Function and Nonstandard Finite<br>- Non-Standard Finite Difference (NSFD)<br>- Fourth-order Runge-Kutta (RK4) method |
| [15] | Botswana South Africa | Predict Growth and Transmission | $S_{SC}IR_E$ | - Partial Rank Correlation Coefficients (PRCC)<br>- Latin Hypercube Sampling (LHS)<br>- Basic Reproduction Number ($R_0$)<br>- Kamgang-Sallet Stability Theorem<br>- Fisher's Transformation |
| [18] | Russia Israel | Predict Growth and Transmission | SIR | - Logarithmic scale<br>- Autoregressive Distributed Lag model (ADL)<br>- Partial Auto-Correlation Function (PACF) |
| [19] | India | Predict Growth and Transmission | SIR | - Generalized Inverse Weibull distribution<br>- Fisher-Tippet (Extreme Value distribution)<br>- Log Normal functions<br>- Levenberg-Marquardt (LM) for curve fitting<br>- Gaussian model |
| [20] | Pakistan | Spread | SIR | - Linear regression |
| [21] | Croatia | Spread | SIR | - Grid search algorithm<br>- Time-series |
| [22] | Germany | Spread | SIR | - Genetic Algorithm (GA)<br>- Particle Swarm Optimizer (PSO)<br>- Grey Wolf Optimizer (GWO) |
| [23] | Iran | Prevalence or Decrease in Spread with respect to other factors | $SIR_D$ | - Growth distribution<br>- Gaussian Function<br>- Logistic Function<br>- Support Vector Machine<br>- Linear regression algorithm |
| [24] | China India Turkey | Prevalence or Decrease in Spread with respect to other factors | $SIR_U$ | - Fractional Riemann-Liouville<br>- Caputo fractional order derivative<br>- Laplace transform and q-HATM series |
| [25] | India | Prevalence or Decrease in Spread with respect to other factors | $SIR_{VD}$ | - Artificial Neural Network-based Adaptive Incremental Learning technique (ANNAIL)<br>- Jacobian Matrix |

The state of the art presented in this paper revealed a set of research works that directly use the SIR model in conjunction with mathematical models to predict the evolution of the COVID-19 pandemic. In research works [14,15,18,19] the SIR model is applied to analyze and predict the growth and transmission of the COVID-19 disease by exploring the incidence of the coronavirus modeled in 10-day intervals. In these approaches, the data of the evolution of COVID-19 in different regions is used to predict the threat through different theoretical metrics.

In [20–22], the spread of COVID-19 is modeled with the SIR model in order to know the future trends and the maximum number of patients who are recovered, deceased, or infected. The research work presented in [3] evaluates the effectiveness of quarantine protocols to limit the spread of COVID-19 using Bayesian networks.

The study of the state of the art revealed some other research works focus on improving the SIR model. These research works add new subcategories to the original states of SIR model, parameterizing the model in order to study the transmission dynamics and the prevention mechanisms. The works in this category analyzed several conditions to be taken into account to explain a trend of prevalence or decrease in the spread of COVID-19, such as deaths [23], unreported cases [24], vaccination [25], as well as individuals who decided to avoid any protection mechanism [15].

The SEIR model, which is a derivation of the SIR model, considers an incubation period in which individuals are carriers of the disease but they do not have symptoms and they cannot infect others. These individuals are in the *Exposed* stated. When this period ends and the individuals can already infect others, they move to the *Infected* state. The basic idea behind the SEIR model is to express change with respect to time and the movement of individuals from one compartment to another; in essence, this is the effect that the spread of the disease has on the population [9].

Table 3 presents research works that combine mathematical approaches with the epidemiological SEIR model. The simple SEIR model is used in [26–28] together with epidemiological data obtained from several official sources in order to produce a predictive analysis of the COVID-19 epidemic outbreak. In [29], a tentative forecast (adding other categories within states) was made taking into account an initial outbreak in Lombardy, Italy. The evolution of COVID-19 was predicted in research works [12,30] by modeling mitigation scenarios through the continuous improvement of detection techniques, the forecasts of hospital demand [31], and the applications of policies for city closures.

**Table 3.** Prediction algorithms using the SEIR epidemiological model.

| Reference | Country | Task Performed | Epidemiological Mathematical Model | Mathematical Prediction Algorithms |
|---|---|---|---|---|
| [12] | India Vietnam Nigeria | Mitigation scenario modeling | SEIR$_{(AQ-I)}$ | - Cumulative distribution function<br>- Probability density function<br>- Lyapunov's stability theorem<br>- Prophylaxis |
| [26] | Colombia | Predictive analysis of the epidemic outbreak | SEIR | - Different Basic Reproduction Numbers ($R_0$) = 1.5, 2.0, 2.4, 2.6 |
| [27] | China | Predictive analysis of the epidemic outbreak | S$_{(in-out)}$ EIR | - Long-Short-Term-Memory (LSTM) model<br>- Ljung-Box (LB) test |
| [28] | Algeria | Predictive analysis of the epidemic outbreak | SEIR | - Basic Reproduction Number ($R_0$) |

**Table 3.** *Cont.*

| Reference | Country | Task Performed | Epidemiological Mathematical Model | Mathematical Prediction Algorithms |
|---|---|---|---|---|
| [29] | Egypt | Predictive analysis of the epidemic outbreak | $SEI_IR_D$ | - Mixed-Integer Optimization<br>- Functions: Normal, Log-normal, Weibull, Beta, Gamma, Burr, Exponential, and Birnbaum-Saunders |
| [30] | USA China | Mitigation scenario modeling | SEIR with a FDE model | - Fractional-Derivative Equations (FDE)<br>- Partial Differential Equation (PDE)<br>- Sigmoid Function<br>- Lagrangian approach<br>- Interaction radius R<br>- Caputo and Riemann Liouville fractional derivatives |
| [31] | México | Mitigation scenario modeling | $SEIR_D$ | - AMA model<br>- Bayesian Uncertainty Quantification<br>- Bayesian Inference |
| [32] | USA | Trending of change in disease dynamics | $SEIR_{DP}$ | - Short history window data<br>- Nonlinear least-squares<br>- Simple Forward Euler method<br>- Runge-Kutta method |
| [33] | Saudi Arabia | Spread modeling | $SI_{DA}R_{TH}E$ | - Ordinary differential equations<br>- Lagrange polynomial<br>- Lyapunov<br>- Atangana-Baleanu fractional derivative in Riemann-Liouville sense<br>- Caputo-Fabrizio fractional derivative<br>- Adams-Bashforth method |

In another research approach [32], historical data was used to find the trend of change in disease dynamics, with the purpose of finding real-time adaptive changes of disease mitigation, commercial activity, and social behavior of the populations.

Finally, in research work [33], a set of eight nonlinear ordinary differential equations were used to model the spread of COVID-19 in a certain population by using the concept of a next-generation matrix in order to derive a basic reproduction number. This number details a possible stability in the equilibrium of the SEIR model with numerical simulations for different values of fractional orders.

*4.3. Prediction with Machine Learning and/or Data Mining Algorithms*

Today, there is growing interest in the use of machine learning techniques to explain and predict the dynamics of COVID-19. These data mining approaches take into account several factors that are related to the evolution of the pandemic in several countries. Table 4 presents research works that use techniques of machine learning and/or data mining for prediction.

**Table 4.** Prediction algorithms using machine learning and/or data mining techniques.

| Reference | Prediction Task Performed | Techniques of Machine Learning or Data Mining |
|---|---|---|
| [3] | Impact of 21-day lockdown on the number of infections | - Bayesian model<br>- Poisson Regression<br>- Monte Carlo in Markov Chain (MCMC) |
| [11] | Initial Period of an Outbreak | - MCMC+GROOMS methodology<br>- Optimized Broyden-Fletcher-Goldfarb-Shanno Polynomial Neural Network (BFGS-PNN) based on fuzzy rules |
| [19] | Forecast in the growth and transmission of the COVID-19 disease exploring its incidence in 10-day intervals | - Generalized Inverse Weibull Distribution<br>- Fisher-Tippett (Extreme Value Distribution)<br>- Levenberg-Marquardt (LM) for curve fitting<br>- Gaussian model |
| [20] | Possible incidence | - Linear regression |
| [21] | Mining data to predict the spread of a pandemic | - MLP with activation functions ReLU<br>- Limited-memory Broyden-Fletcher-Goldfarb-Shanno Algorithm |
| [22,34] | Epidemic forecast | - Multi-Layer Perceptron ANN<br>- Adaptative Neuro Fuzzy Inference System<br>- Radial Basis Neural Network<br>- Ensemble Empirical Mode Decomposition<br>- Genetic Algorithms<br>- Particle Swarm Optimization<br>- Grew Wolf Optimizer. |
| [23] | Predict the peaks of an outbreak | - Support Vector Machine |
| [25] | Transmission dynamics and the prevention mechanism of COVID-19 | - Adaptative Incremental Learning-based ANN<br>- ANN and Jacobian<br>- Multi-Layer Perceptron ANN |
| [27] | Predict the peaks and amplitude of the epidemic | - ANN |
| [32] | Predict specific risk | - LSTM-based ANN |
| [35] |  | - ANN Bayesian |
| [36] | Evaluate the accumulated data of confirmed cases of COVID-19 in Egypt | - ANN Nonlinear Autoregression |
| [37] | Modeling of cumulative data of confirmed cases from 8 European countries | - Nonlinear Auto Regression Neural Network<br>- Autoregressive Integrated Moving Average<br>- Long Short-Term Memory |

**Table 4.** *Cont.*

| Reference | Prediction Task Performed | Techniques of Machine Learning or Data Mining |
|---|---|---|
| [38] | Assess the effectiveness of mitigation measures | - ANN Recurrent |
| [39] | | - Naïve method<br>- Holt-Winters method<br>- Autoregressive Integrated Moving Average |
| [40] | Predict the evolution of the COVID-19 pandemic | - MLP with Logistic Regression algorithms<br>- Fuzzy<br>- Autoregressive Integrated Moving Average |
| [41] | | - ANN based LSTM |
| [42] | Predict recovery of infected patients | - Logistic Regression<br>- Support Vector Machine<br>- Decision Tree<br>- Naïve Bayes<br>- Random Forest<br>- K-Nearest Neighbor |
| [43] | Use of geographic and demographic data to predict case severity and possible recovery or death | - Random Forest<br>- AdaBoost Algorithm |

One of the research projects that analyzed and predicted the growth of the pandemic using machine learning is the work presented in [19], where the authors applied an improved mathematical model based on machine learning techniques (Generalized Inverse Weibull Distribution) within a SIR model to forecast the potential threat of COVID-19 in countries around the world.

An Artificial Neural Network (ANN) technique that is suitable for the prognosis of the epidemic is presented in research works [22,34]. These works compare three different techniques: (a) Multi-Layer Perceptron (MLP), (b) the Adaptive Network-based Fuzzy Inference System (ANFIS), and (c) the Extender Empirical Mode Decomposition (EEMD).

In these papers, an analysis is also made using the results obtained from tools such as Genetic algorithms, Particle Swarm Optimization and Grew Wolf Optimizer. The paper results reveals that comparison between analytical and machine learning model indicates that the Multi-Layer Perceptron delivered the most accurate results.

In [3], a Bayesian extension of the eSIR (extender SIR) model was used together with the Poisson Regression Model and Markov-Chain Monte Carlo (MCMC) to study the short- and long-term impact of an initial 21-day lockdown on the total number of COVID-19 infections in India compared to other, less severe nonpharmaceutical interventions.

In [17], the authors used the Composite Monte-Carlo model (CMCM) and its proposed methodology, which is called Group of Optimized and Multisource Selection (GROOMS). This model finds the best performing deterministic forecasting algorithm. A case study of the recent novel coronavirus epidemic was used as an example in demonstrating the efficacy of GROOMS+CMCM. The paper concludes that the proposed approach can be successfully used in the initial period of disease outbreaks, when data is sparse and full of uncertainty.

In the Mexican context, two mathematical models (Gompertz model and Logistic model) and one computational model (inverse Artificial Neural Network) were applied in [35] to estimate the cases of infection of COVID-19 through confirmed cases, where the

main data mining tool was a Bayesian Artificial Neural Network. The results of the paper shown a good fit between the estimated and the observed data on total confirmed cases, given as a result a better estimation of Artificial Neural Network models.

In [36], the authors propose the use of non-linear autoregressive of Artificial Neural Network (NARANN) to evaluate the data accumulated of confirmed COVID-19 cases in Egypt and compare the results with autoregressive integrated moving average (ARIMA). The results of the paper indicated that NARANN showed a better performance compared with ARIMA based on different statistical criteria.

In [37], the accumulated data modeling of confirmed cases from eight different European countries (Denmark, Belgium, Germany, France, United Kingdom, Finland, Switzerland and Turkey) was performed using a Nonlinear Autoregression Artificial Neural Network (NARNN) added to the Self-Regressive Integrated Moving Average Model (ARIMA) with Long-Short Temporal Memory (LSTM). According to the results of the study, LSTM was found the most accurate model to predict the evolution of the pandemic. This Long-Short Temporal Memory (LSTM) model was also used in [38] to understand and evaluate the effectiveness of mitigation measures in the COVID-19 propagation. According to the authors, the LSTM model can learn from the cumulative rise in COVID-19 confirmed cases and deaths and provide valuable insights on how well mitigation measures are working quantitatively in terms of the rate of infections and deaths.

Second-generation Artificial Neural Networks are used in research work [21] as the main data mining tool to predict the spread of a pandemic; this work also uses a Multilayer Perceptron (MLP) with ReLU activation functions and LBFGS optimization methods to improve the prediction efficiency. In the research work [39], a similar approach was followed, but other algorithms, such as Naive Method, Holt-Winters Method and ARIMA were used.

The authors of research work [40] explore the use of Multilayer Perceptron (MLP) in conjunction with Logistic Regression, Fuzzy, and ARIMA algorithms for providing a method to predict the evolution of the COVID-19 pandemic. The results indicated that implement hybrid Artificial Neural Networks using data transformation techniques based on improved algorithms, combining forecast models, and using technological platforms enhance the learning and generalization of ANN in forecasting epidemics.

In research works [32,41] an LSTM-based Artificial Neural Network guided by Bayesian optimization was used to predict the specific risk of COVID-19. In [20], the authors propose the use of Linear Regression to predict the possible future incidence of cases in Pakistan. In [23], the same technique is used to analyze a data set that includes the daily number of death cases, confirmed cases, and recovered cases in Iran. This approach also uses Vector Support Machines and Gaussian functions to predict the peaks of an outbreak. The authors of this paper concludes that Gaussian functions are the best method to predict the outbreak because this function has enough efficiency to estimate the curve, peak and the end of the current COVID-19 pandemic.

In the study of the transmission dynamics and the prevention mechanism of COVID-19 presented in [25], parameters were estimated with an analytical model guided by the flow of data using an Artificial Neural Network-based Adaptive Incremental Learning technique (ANNAIL). The authors established that the proposed algorithm can eliminate the need to rebuild the model from scratch every time a new training data set is received. The main advantage of this model is its capacity to support real-time online incremental learning which enables the algorithm to incorporate new characteristics of the pandemic, for example change in transmission dynamics due to mutations in virus, change in prevention mechanisms or government policies, etc.

In [42], the authors developed five data mining models for predicting the recovery of infected patients using a set of epidemiological data from patients with COVID-19 from South Korea. In this work, the Random Forest technique obtained the least uncertainty compared to the other models.

An Artificial Neural Network was incorporated in [27] to a SEIR epidemiological model to predict the peaks and sizes of the COVID-19 epidemic. The authors mentioned that the dynamic SEIR model was effective to predict the peaks and sizes, however, the model did not take into account factors that could increase the number of confirmed case numbers, such as diagnostic capacity or seasonal influences.

In [43], the geographic, health and demographic data of patients with COVID-19 are used to predict the severity of the case and possible recovery or death. The techniques used in this work are Random Forest Model driven by the AdaBoost algorithm in conjunction with an Algorithm Grid Search.

### 4.4. Performance Measurement or Evaluation

The results obtained from a model must be measured and evaluated in order to determine its deficiencies and/or performance, and to compare it with other state-of-the-art papers in the same field. The evaluation measures of the internal components of a model can be used to evaluate the technologies used in each component as a function of its design parameters [44]. Table 5 shows the evaluation tools applied in each one of research works considered in this paper.

**Table 5.** Performance or evaluation measures.

| References | Performance Measurement or Evaluation. |
|:---:|:---:|
| [17,20,36–39] | Root Mean Square Error (RMSE) |
| [18,19,21,34–36] | Coefficient of Determination (R2) |
| [19,22,27,34,37] | Mean Square Error (MSE) |
| [19,37] | Mean Absolute Percentage Error (MAPE) |
| [20,36] | Mean Absolute Error (MAE) |
| [21] | Cross-validation using K-fold algorithm with 5-folds |
| [22] | Correlation Coefficient |
| [31] | Runge-Kutta method |
| [34] | MinMaxScaler |
| [36] | Deviation Ratio (DR) Coefficient of Residual Mass (CRM) |
| [37] | Peak Signal-to-Noise Ratio (PSNR) Normalized Root-Mean-Square Error (NRMSE) Symmetric Mean Absolute Percentage Error (SMAPE) |
| [40] | Sum of Squared Error (SSE) |
| [41] | Bayesian optimization |
| [42,43] | Accuracy |
| [43] | Precision Recall F1 Score |

*4.5. COVID-19 Datasets*

The input data is one of the relevant aspects that is very relevant in the successful application of the novel approaches to predict the dynamics of the COVID-19 pandemic. This is the reason why one of the relevant contributions of this paper is the determination of the sources of the information used in each one of the researches works analyzed in this study. The datasets used in each approach are presented in Table 6.

**Table 6.** COVID-19 Dataset descriptions with available resources.

| Reference | Description | Data Sources |
|:---:|:---:|:---:|
| [3] | Official Data of India | covid19india.org (accessed on 1 May 2021) |
| [3,18,21,34] | Johns Hopkins University Center for Systems Science and Engineering (JHU CSSE) ESRI Living Atlas and the Johns Hopkins University Applied Physics Laboratory (JHU APL) | https://coronavirus.jhu.edu/map.html https://seandavi.github.io/sars2pack/reference/jhu_data.html (accessed on 1 May 2021) |
| [17] | Data from the Chinese Center for Disease Control and Prevention (CDCP) | http://www.chinacdc.cn/en/ (accessed on 1 May 2021) |
| [18,23,39] | OMS Global Data Set. | https://data.humdata.org/dataset/coronavirus-covid-19-cases-and-deaths, 2020 (accessed on 1 May 2021) |
| [19] | Global dataset, with some interactive charts. | Our World in Data de Hannah Ritchie2 https://ourworldindata.org/coronavirus-source-data https://collaboration.coraltele.com/covid/ (accessed on 1 May 2021) |
| [20] | Global data set of time series, confirmed cases, recovered and deaths. | Humandata.org. (accessed on 1 May 2021) |
| [22] | Global data for Italy, Germany, Iran, USA, and China | https://www.worldometers.info/coronavirus/country (accessed on 1 May 2021) |
| [26] | Secretaría de Salud Municipal de Cali. Departamento Administrativo Nacional de Estadística. | https://www.cali.gov.co/salud/publicaciones/152840/boletines-epidemiologicos/ https://www.dane.gov.co/index.php/estadisticas-por-tema/demografia-y-poblacion/proyecciones-de-poblacion (accessed on 1 May 2021) |
| [27] | China National Health Commission. | http://www.nhc.gov.cn/xcs/yqtb/list_gzbd.shtml (accessed on 1 May 2021) |
| [29] | Emphasis was placed on the basic reproduction number $R_0$. | COVID-19 Community Mobility Reports released by Google |
| [31,35] | Secretaría de Salud de México | https://www.gob.mx/salud/documentos/coronavirus-covid-19-comunicado-tecnico-diario-238449 (accessed on 1 May 2021) https://coronavirus.conacyt.mx/proyectos/ama.html (accessed on 1 May 2021) |

**Table 6.** *Cont.*

| Reference | Description | Data Sources |
| :---: | :---: | :---: |
| [36] | Official data from the Egyptian Ministry of Health and Population. | https://www.gavi.org/covid19?gclid=CjwKCAjw7J6 EBhBDEiwA5UUM2j8GID2GCD588LbiMzU2L1 ragN06l1Ct7kSNuxbKX0AuiLUsexiKDhoCxI4 QAvD_BwE (accessed on 1 May 2021) |
| [37] | European Center for Disease Prevention and Control. | https://www.ecdc.europa.eu/en/publications-data/ download-todays-data-geographic-distribution-covid-19-cases-worldwide (accessed on 1 May 2021) |
| [42] | Korea Centers for Disease and Prevention (KCDC). | http://www.kdca.go.kr/index.es?sid=a3 (accessed on 1 May 2021) |
| [43] | Official Data of South Africa. | https://www.kaggle.com/sudalairajkumar/novel-corona-virus-2019-dataset/ https://github.com/Atharva-Peshkar/Covid-19 -Patient-Health-Analytics (accessed on 1 May 2021) |

## 5. Discussion of the State of the Art

This paper have pointed out the importance of data mining correlation in the analysis of epidemiological models since the two fields share similar tasks. These include:

1.  Explaining and predicting the dynamics of a disease.
2.  Discovering patterns or extracting relevant information.
3.  Taking into account that mathematical models (like some data mining tools) has several variations in their complexity.
4.  Studying the pandemic evolution in the short and medium term
5.  Predicting values based on historical data.

Our study found that the SEIR model can be effectively adapted to determine the behavioral dynamics of a pandemic. However, it is important to take into account the previous knowledge of the pandemic to improve the efficiency of the model by modifying its parameters and number of inferred variables.

The use of data mining techniques is relevant in current pandemic scenario, since data mining techniques allow the appropriated tunning of epidemiological models to effectively match the real evolution of the pandemic.

Several papers have demonstrated the advantages of using Artificial Neural Networks in the resolution of the risk prediction for COVID-19 and also in the propagations of the pandemic [21,39–41], even in situations with incomplete data [17]. Other papers revealed the usefulness of Artificial Neural Networks in the estimation and prognosis of confirmed cases of the disease [22,34,35] and also in the evaluation and application of containment measures [20,25,36–38].

This statement should be interpreted with caution. Even though artificial neural networks have processed information and detected non-linearities in systems with time series, these results become another source of complex problems since the variations in the initial conditions of a mathematical epidemiological model can involve large differences in future behavior of diseases, making it impossible to make an accurate prediction in the short term [3].

Similar results were obtained in the study of the evaluation metrics. The Root of the Mean Square Error (RMSE), the Mean Square Error (MSE), and the coefficient of determination (R2) have been frequently used in the study of epidemiological models. because they allow the variation of the resulting values to be evaluated and the prediction

or testing of future results. Since hybrid models can be useful in the identification of that help in the prediction and evaluation of the spread of COVID-19, they can be used in managing the measures to minimize the effects of epidemic diseases. One of the aspects that needs to be analyzed in future work is the quality of data obtained from different countries. There is still uncertainty as to its veracity, especially the data collected during the onset of the pandemic, since it is often generated incompletely or with some delays. This could cause noise at the time of feeding the different machine learning models, which make it difficult to obtain an accurate prediction of the behavior of the virus or to provide appropriate measures to prevent its spread in the population.

The study also reveals that some countries do not have enough serological studies in their population, causing the interpretation of the results to be ambiguous.

One of the findings of our review is that most of the research papers that were analyzed used a combination of different data mining techniques to improve a specific epidemiological model. Only in few cases an epidemiological model was combined with only one data mining technique.

Another of the review's findings was that data mining algorithms were used for very different purposes in COVID-19 context: predicting the transmission of the disease, predicting the peaks of the epidemic, predicting the dynamics of the spread of the virus, measuring the effectiveness of the measures to fight the COVID-19, predicting the recovery of infected patients, etc. This large number of topics reflects the importance of mining techniques at different stages of the pandemic.

Figure 1 graphically synthesizes the correlation among COVID-19 concepts we have found as result of the review carried in this paper. The figure detail the task of detection, diagnosis, forecasting of the pandemic and the join with data mining and machine learning algorithms that extend current epidemiological models. The figure synthetizes the different categories of data mining techniques applied to prediction tasks in COVID-19 context.

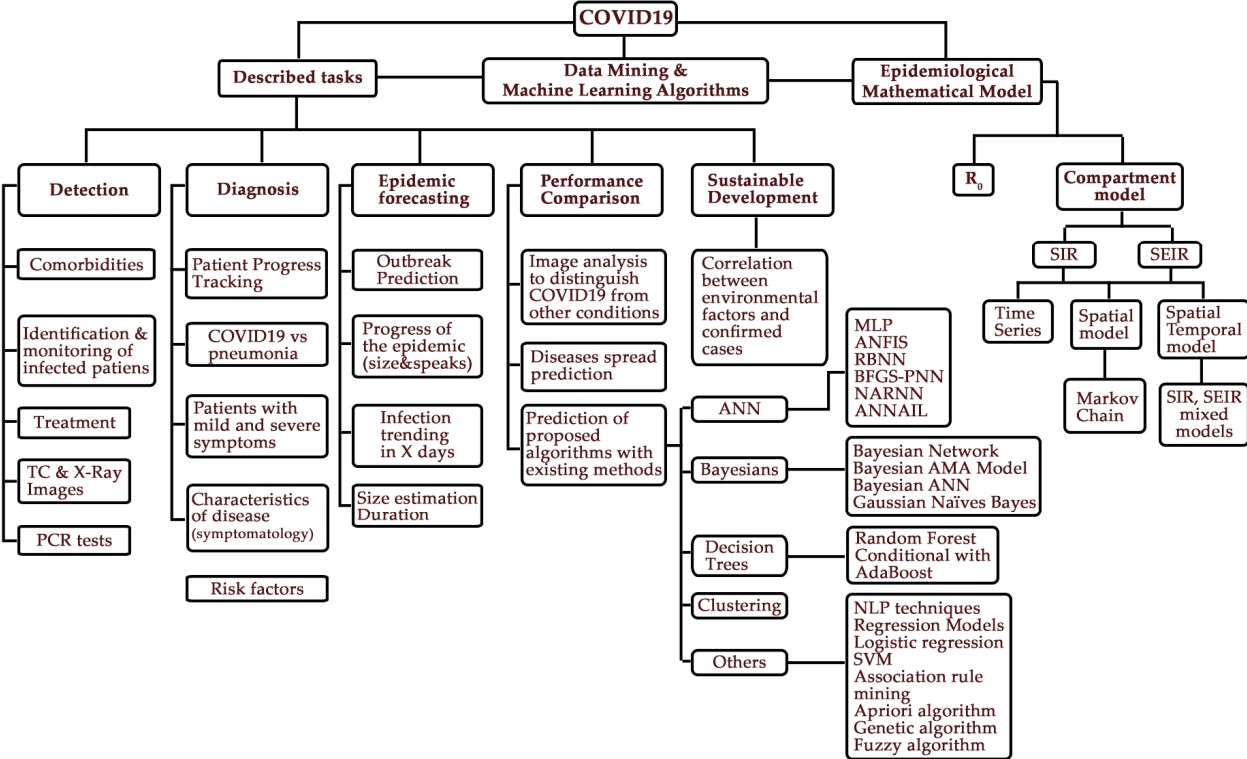

**Figure 1.** Tree diagram with main concepts found during literature review.

## 6. Conclusions and Future Work

This research analyzed 32 papers, seven books and three surveys that are related to the prediction of the COVID-19 pandemic using data mining algorithms and mathematical epidemiological models. The study revealed that there is great interest, in the scientific community, in the use of computational models combined with epidemiological algorithms to face the great challenge of predicting the evolution of the current COVID-19 pandemic. The new variants of the disease complicate the prediction tasks however it is possible to establish that the models presented in this paper are promising in predicting the behavior of susceptible, infected and recovered individuals.

Of the trends in the area is the use of hybrid models that combine the advantages of several IA algorithms. We have only been able to find a few research papers that use a single model to predict the evolution of the pandemic.

We are currently defined a novel research work is being defined to use data mining techniques for extending the current SEIR epidemiological model used in Mexico with additional factors such as mobility, economic development, and pollution in Mexican cities.

**Author Contributions:** Conceptualization, H.E.-E.; methodology, K.V.C.-M., H.E.-E., A.M.-R., Y.H.-P. and J.O.-H.; investigation, K.V.C.-M. and H.E.-E. and A.M.-R.; writing—original draft, K.V.C.-M.and H.E.-E. and A.M.-R. and Y.H.-P.; writing—review and editing, and Y.H.-P. and J.O.-H. All authors have read and agreed to the published version of the manuscript.

**Funding:** This research received no external funding.

**Institutional Review Board Statement:** Not applicable.

**Informed Consent Statement:** Not applicable.

**Data Availability Statement:** All Datasets used for the research works analyzed in this paper were reported in Table 6: COVID-19 Dataset descriptions with available resources.

**Conflicts of Interest:** The authors declare no conflict of interest.

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
