# Peer review of "The State of the Art of Data Mining Algorithms for Predicting the COVID-19 Pandemic"

_axioms, doi:10.3390/axioms11050242_

Round 1

Reviewer 1 Report

This is a review article. Existing research and the author's opinion on the problems have been thoroughly studied.
I only have remarks on the structuring of the article. Paragraphs 2.1, 2.2 and 3.2.1 do not need to be separated because they are small in size.
In addition, paragraph 4 could be extended with clear conclusions from the study.

Author Response

Comment 1. I only have remarks on the structuring of the article. Paragraphs 2.1, 2.2 and 3.2.1 do not need to be separated because they are small in size.

Answer:  To solve this kind comment, we have eliminated sections 2.1 and 2.2. This sections was included as bullets as follow:

The stability of an epidemiological model can be defined by its equilibrium points, which are usually defined in terms of the Basic Reproduction Number . Following, we define the concepts of equilibrium in epidemiological systems and the basic reproduction number:

  • Equilibrium epidemiological systems. One of the important objectives in the study of a disease is to know if it will persist in a specific population, or if this disease gradually disappears until its eradication [9]. There are two types of equilibrium: Disease-free and Endemic. Disease-free Equilibrium: In this type, the proportion of exposed and infectious individuals is zero so that the disease is eradicated [9] [12], that is, this situation occurs when the infectious subpopulations are canceled [13]. Endemic Equilibrium: In this type, the proportion of exposed and infectious individuals is not zero [14]; however, the capacity of the system is stable enough to remain in equilibrium even with small disturbances. These small disturbances cause the epidemiological models to begin to evolve away from a state of infection [9]. In this case, the disease cannot be totally eradicated but remains in the population. The equilibrium in epidemiological systems has been thoroughly analyzed in the research works [12] [14] and [15] where they simulate preventive measures, containment, and the possible impact of a pandemic in the population.
  • Basic Reproduction Number. In an epidemic, a very important parameter is the Basic Reproduction Number , it allows us to distinguish between an epidemic state (when the pathological process exceeds the expected contagions and the geographical limits) and an endemic state (when the pathological process remains stationary for an extended period of time). The Basic Reproduction Number is essential to be able to understand the nature of the different diseases and their temporal evolutions [13]. The is defined as the mean number of secondary infections that occur when an infectious individual is introduced into a su ceptible population. It can be represented as  [9], that is, the number of individuals to be infected from patient zero. In order to eradicate an infection, it is necessary to reduce the  below unity [13]. This is sometimes accomplished through immunization programs, which have the effect of transferring members of the susceptible class (S) to the recovered class (R). The smaller the , the slower the epidemic will evolve. In practice, for a specific real epidemic, observing the epidemic allows us to measure  and, them estimate [17].

Regarding the section 3.2.1 this was renumbered, this is because this section provides information for the metrics used in mathematical models and data mining techniques therefore, it is incorrect to place this section inside section 3.2. Currently this section was renumbered as 4.4Performance measurement or evaluation because we have added a new section 2. Related works.

Comment 2: In addition, paragraph 4 could be extended with clear conclusions from the study.

Answer. Thanks to this kind reviewer comment we have improved section 4. It is important to point out that because we have added a new Related work Section, the Discussion of the state of the art has been renumbered as section 5. The findings that were defined in the paper are the following:

One of the findings of our review is that most of the research papers that were analyzed used a combination of different data mining techniques to improve a specific epidemiological model. Only in few cases an epidemiological model was combined with only one data mining technique.

Another of the review's findings was that data mining algorithms were used for very different purposes in COVID-1 context: predicting the transmission of the disease, predicting the peaks of the epidemic, predicting the dynamics of the spread of the virus, measuring the effectiveness of the measures to fight the COVID-19, predicting the recovery of infected patients, etc. This large number of topics reflects the importance of mining techniques at different stages of the pandemic.

Figure 1 graphically synthesizes the correlation among COVID-19 concepts we have found as result of the review carried in this paper. The figure detail the task of detection, diagnosis, forecasting of the pandemic and the join with data mining and machine learning algorithms that extend current epidemiological models. The figure synthetizes the different categories of data mining techniques applied to prediction tasks in COVID-19 context.

Reviewer 2 Report

Kindly see the attached file.

Author Response

Comment 1. Introduction part is rather short and lacks problem introduction, like why data mining algorithms are important regarding COVID-19 prediction?

Answer:  To solve this kind comment, the introduction section has been improved by including information about the relevance of data mining algorithms in the context of COVID-19. The following paragraph was included in the paper:

The relevance of data mining techniques in the context of COVID could be explained with their ability to find hidden patterns using large databases for a very new disease without previous knowledge about its evolution in time. In the case of COVID data mining data mining techniques are useful to extract meaningful information from complex raw data providing multiple benefits in the healthcare sector [6]. Safdari [7] also mentioned the significant impact of data mining techniques on selecting the most effective techniques in pandemic studies by helping the researchers to reveal the unknown character of a new pandemic and the next possible pandemics. This paper aims to highlight the relevance of the integration of mathematical epidemiological models with data mining techniques and/or machine learning algorithms in predicting the evolution of the COVID19 pandemic. The state-of-the-art review covers 32 research articles, seven books, and three surveys that one related to this topic.

Comment 2. Authors need to add closely related works, similar to their study.

Answer: This was a very relevant comment for our paper. To solve this issue, we have included a new related work section with 3 research works that share the objective of exploring the use of data mining techniques in the prediction of COVID-19. We have defined the objective of that reviews and also we have indicated the differences of the related works with the research work presented in our paper.

Comment 3. Add the study selection method, including inclusion and exclusion criteria.

Answer: Thanks a lot by this comment that enables to improve the paper including section 4.1 were we explained the approach used in our paper to select the papers. The section included is the following:

4.1 Approach used for the papers selection

It is important to point out that current version of paper of the paper doesn’t pretend to be a systematic review of the literature of data mining techniques applied in the prediction of COVID evolution. This paper has been defined as a preliminary scoping search of relevant works. However some of the steps of a systematic review has been followed in our paper:

The questions to answer in this search were: What are the different data mining techniques that have been used to predict the behavior of the COVID-19?  What specific epidemiological model is used in that papers? What are the metrics used to measure the precision of the proposed data mining techniques? What are the data sources of the analyzed research works?

The inclusion criteria was to consider all works using mathematical algorithms and data mining techniques combined with a specific epidemiological model (SEIR, SIR,eSIR, etc) to intend to predict the pandemic behavior of COVID-19

The exclusion criteria was to eliminate all works using data mining techniques for other not pandemic diseases and also eliminate of the review all papers that do not have the basis of an epidemiological model (SEIR, SIR,eSIR, etc) to be combined with methodological models or data mining techniques.

This inclusion and exclusion criteria are the reason why this paper does not consider the use of alternative techniques to detect/predict the COVID-19, such as sentiments analysis based on natural language processing techniques, images recognition for COVID-19 detection, use of sensor for preliminary diagnosis, etc. This paper is focus on mathematical models and data mining techniques that extend/improve the current epidemiological model proposed by health organizations.

Following, we present the research works for predicting COVID-19 with epidemiological models using mathematical algorithms and the research work for predicting COVID-19 with machine learning and/or data mining algorithms.

Comment 4. Several recent works related to COVID-19 are not included in the study, (Analyzing Sentiments for COVID-19 Vaccination Using Lexicon-Based Approach with LSTM-GRNN Ensemble: A Case Study on Worldwide Tweets; COVINet: a convolutional neural network approach for predicting COVID-19 from chest X-ray images; Deep Learning Based Early Detection Framework for Preliminary Diagnosis of COVID-19 via Onboard Smartphone Sensors; Prediction models for COVID-19:Integrating Age Groups, Gender, and Underlying Conditions)

Answer: In order to give solution to this point we have included the inclusion and exclusion criteria that enable us to select the analyzed papers. The following explanation was included in the paper:

The exclusion criteria was to eliminate all works using data mining techniques for other not pandemic diseases and also eliminate of the review all papers that do not have the basis of an epidemiological model (SEIR, SIR,eSIR, etc) to be combined with methodological models or data mining techniques.

This inclusion and exclusion criteria are the reason why this paper does not consider the use of alternative techniques to detect/predict the COVID-19, such as sentiments analysis based on natural language processing techniques, images recognition for COVID-19 detection, use of sensor for preliminary diagnosis, etc. This paper is focus on mathematical models and data mining techniques that extend/improve the current epidemiological model proposed by health organizations.

Comment 5. Notations used in the text looks like copy-pasted as images, use notations properly

Answer: To solve this kind comment we have reviewed all the notations to include this as part of the text.

Round 2

Reviewer 2 Report

My comments are resolved, except for using the proper mathematical notations. Math symbols are used as images and they look blurry in the paper. Authors are advised to fix this problem.

Author Response

Thanks a lot by the kind comment about the mathematical notations. We have solve this issues by eliminating the images of the symbols and by placing the notation directly in the text, which has improving the quality of the mathematical notations.